# Field Assessment of Downed Timber Strength Deterioration Rate and Wood Quality Using Acoustic Technologies

Munkaila Musah [1,2,*] , Javier Hernandez Diaz [1,2], Abiodun Oluseun Alawode [1,2] , Tom Gallagher [1], Maria Soledad Peresin [1,2], Dana Mitchell [3], Mathew Smidt [3] and Brian Via [1,2]

1  College of Forestry and Wildlife Sciences, Auburn University, Auburn, AL 36849, USA; jah0189@auburn.edu (J.H.D.); aza0236@auburn.edu (A.O.A.); gallatv@auburn.edu (T.G.); soledad.peresin@auburn.edu (M.S.P.); brianvia@auburn.edu (B.V.)
2  Forest Products Development Center, Auburn University, 520 Devall Drive, Auburn, AL 36849, USA
3  Southern Research Station, USDA Forest Service, 521 Devall Drive, Auburn, AL 36849, USA; dana.mitchell@usda.gov (D.M.); mathew.smidt@usda.gov (M.S.)
*  Correspondence: mzm0263@auburn.edu; Tel.: +1-9062313409

**Abstract:** Hurricane and tornado events cause significant damage to high-value timber in the United States each year. Forest managers and landowners are keenly interested in finding solutions to salvage and repurpose these downed timbers before they cause pest infestations and fire outbreaks, completely losing their value or increasing processing costs. To better understand the wood quality of the downed timber, we used acoustic waves techniques as a nondestructive testing approach to assess the wood degradation rate of downed trees and determine the extent of fracture and voids in the damaged regions. We periodically monitored the acoustic velocity of the downed trees for 12 consecutive months using a time of flight (TOF) acoustic method. Acoustic measurements were conducted using three different techniques—longitudinal, transverse, and off-set methods. Wood density, age, and the diameter at breast height (dbh) class measurement for southern timber (chip-n-saw for dbh 8"–11" and sawtimber with dbh 12" and up) were used as the predictive parameters of the downed trees. The results indicated positive relationships between dbh class, stand age, and acoustic velocity measurement ($R^2 > 65\%$). The TOF acoustic velocity was indicated to potentially separate higher-stiffness timber from lower-stiffness timber in a hurricane event for structural or non-structural applications. The regression coefficient from the repeated measurements indicated that both age and diameter class strongly impacted the acoustic properties of the downed trees (*p*-value $\leq 0.001$). The sawtimber dbh class recorded a higher acoustic velocity compared to the chip-n-saw type. Fracture, voids, and massive decay in downed trees were detected beyond the visible inspection, features that often are identified by loggers in lower quality wood; however, TOF showed a weak response in picking up incremental deterioration due to changes in specific environmental factors that affected acoustic readings. This study showed that acoustic wave methods could potentially be used as a field evaluation tool for assessing the quality of downed trees.

**Keywords:** acoustic velocity; dbh class; downed timber; non-destructive evaluation; wood stiffness

## 1. Introduction

Tropical storms and hurricanes perpetually strike the United States yearly, causing devastation, loss of lives and property, and continually generating a huge amount of downed timber [1–3]. Based on the variations in weather patterns, the Fifth Assessment Report of the Intergovernmental Panel on Climate Change (IPCC) anticipated an increase in the intensity of these storms [4,5], which strike the southeast of the country at least twice a year [6]. More storms result in more timber damage and the generation of surges of downed timber [7–9]. This generated downed timber not only results in financial losses of the timber, a material that can cost landowners over USD 1 billion [10], but poses a high risk to the forest ecosystem as a source of microbial and insect manifestation [11], potential fuel

for wildfires [12], wildlife habitat destruction [13], and an increase in operational delays and cost in harvesting system designs [14]. These costs and dangers also come with the loss of carbon sequestration activity by standing trees, which consequently affects atmospheric carbon dioxide associated with global warming [15]. Carbon loss with degradation may also impact a landowner's financial bottom line, as carbon becomes a tradeable commodity [16].

Despite the risks, challenges, and cost of harvesting and salvaging wood, timber can be largely downgraded to relatively low-value applications, such as pulp mill furnishings that mostly supply both the sawlog and the wood pellet industries [17,18]. The demotion of this potentially expensive timber for low-grade applications is mostly due to its economic values [19,20]. When more trees are salvaged in the event of major windstorms, hurricanes, cyclones, and tornados [21–23], mills and traditional markets can rapidly become overwhelmed by a surge in volume, driving down the market prices of timber [24–26]. A study by Prestemon and Holmes [27] estimated that the value of timber losses from Katrina and Rita in the affected states of Alabama, Louisiana, Mississippi, and Texas totaled USD 2.25 billion. Some of these losses can be reduced by the adoption of a holistic mitigation approach of identifying alternative strategies for coping with these storms, including the understanding of downed timber quality with time, as well as finding a new use for the salvaged downed timber.

The need to create new products and provide precise information on timber supply, salvage supply, and demand functional parameters depends on the understanding the downed timber as a material based on the assessment of damage, moisture dynamics, decay, and the deterioration of downed timber mechanical properties with time. Little attention is given to downed timber quality after storm damages. Relatively small trees are easily categorized as pulpwood, and rarer, matured, larger tress, whose quality can be assessed by conventional destructive methods, are also easily sorted, leaving the focus on chip-n-saw and saw logs ranging from 12″ to 20″ dbh. Several studies have shown that nondestructive instruments can directly and/or indirectly measure the intrinsic wood quality of small wood samples, logs, or trees [28–30], including the use of an easy and simple-to-operate, non-destructive sound-wave propagation (FaKopp acoustic technologies) on small wood samples, logs and standing trees [31]. The assessment of the acoustic properties of downed trees is important because their stems are harvested without any knowledge of the internal stiffness quality, which has a significant effect on the performance of the material.

Techniques for the non-destructive evaluation (NDE) of trees and logs using sound-wave propagation have been established as reliable predictors of timber quality in the forest industry. These acoustic technologies are also an important tool for log segregation and value recovery, as they can be used to assess the important intrinsic properties of wood, such as stiffness and bending strength, prior to the processing stage [32–34]. Such techniques, therefore, present an opportunity to increase our knowledge of the quality of downed timber for salvaging and conversion into new products such as CLT, OSB, and plastic composites [35,36].

The assessment of the quality of downed tree materials has become a crucial process in the operational value chain, as landowners and the wood processing industry are under increasing economic pressure to maximize extracted values after storm damage. Hence, to address the hindrances of storm-damaged downed timber and help landowners to make decisions regarding the challenges of downed timber, this study used non-destructive approaches such as sound-wave propagation (TOF) to predict the speed of sound with time and evaluate the stiffness and degradation rate of downed trees. Three different measuring approaches were used: the longitudinal, transverse, and off-set or opposite-face methods. Wood density, age, and the diameter at breast height (dbh) class measurement for southeastern timber (chip-n-saw for dbh 8″–11″ and sawtimber with dbh 12″ and up) were used as the predictive parameters of the downed trees [37]. Additionally, the TOF was used to determine fracture/voids in downed timber to enable the prediction of storm-damaged trees. The categories of decay assessment were based on visual assessment of downed

timber in terms of factors, such as splits, bark slippage, blue stains, rots, etc., as well as models of decay as a function of moisture change and tree time of flight (TOF).

## 2. Materials and Methods

### 2.1. Downed Timber Sites and Measurement

The study was conducted at the Solon Dixon Forestry Education Center in Covington County, Alabama (latitude 31°08′51.9″ N and longitude 86°41′54.4″ W). The site in Covington County is entirely in the lower coastal plain physiographic province, with broad, gentle ridgetop sloping [38]. The elevation ranges from 100 feet above mean sea level in the southern part of the county to about 450 feet in the northeastern region. The total annual precipitation for 2020/2021 was 145.767 mm, with an average relative humidity of 85% in midafternoons. The rotation length for the loblolly pine is 30 to 35 years. Stands at the center may be naturally or artificially regenerated following harvest. The average stand size is around 25 acres, with an irregular shape and age between adjoining stands targeted between 5 to 10 years. In the first part of the study, we randomly selected and cut down 30 fresh loblolly pine trees at three different sites based on their ages to mimic hurricane-downed trees as a pilot study. Fifteen trees were selected for the first site, the second site had ten trees, and the third site had five trees (15, 30, and 40 years old, respectively). The loblolly pine (*Pinus taeda* L.) plantation at the site had 8 × 6 ft spacing. All stands were established with rows facing east–west direction, with a highway road separating site 3 from sites 1 and 2. Each tree had 90 total measuring points at a time (30 each for transverse, longitudinal, and offset methods), consisting of 6 replicates at 5 marked locations along the length of the tree, starting at 304.8 mm from the butt of the downed tree at an interval distance of 1400 mm up to the top of the tree. Over 8000 measurements were carried out for all 30 trees bi-weekly for seven months and monthly afterward to complete a year. The data gathered from the tree and specimens of the same tree were collated. The average was used to represent the tree, and those from the same range of diameter classes were pooled together to represent the tree diameter class.

### 2.2. Stress Wave Estimation Procedure

The simple and easy-to-use acoustic tool Fakopp 1D Microsecond Timer (Fakopp Enterprise, Agfalva, Hungary) [31] based on the popular time of flight (TOF) technique [39] was used to measure stress wave velocity, which that was transmitted and received via a probe inserted into the sapwood of each tree at a 45-degree angle, pointed towards each other at opposite ends [33,39,40]. The transverse ($V_{TSWV}$), longitudinal ($V_{LSWV}$), and the opposite phase (off-set) ($V_{OSWV}$) methods were used in this study (Figure 1). The transverse probes were at the same point on the opposite sides of the tree with the probe distance (D) based on the diameter and the circumference of the tree: ($D = \pi d/2$).

The probes were vertically spaced at 1400 mm apart on the same side of the tree for the longitudinal measurement, allowing for redundant measurements of longitudinal stress wave velocity to occur at the same location of each tree ($D = 1.4$). The opposite phase or off-set measurement combined the transverse and longitudinal measurement approach where the probes were placed at opposite ends of the diameter and circumference of the tree, with 1 probe at a distance of 1.4 m in relation to the tree circumference, described as the circumferential opposite-face [41].

The Fakopp 1D Microsecond timer measured the time of flight, and TOF (microseconds μs) measurement was taken for the leading edge of a sound wave, with a velocity allowing it to travel from the sending probe to the receiving probe. The TOF which was an unbound material is governed by a three-dimensional longitudinal wave, the Rayleigh wave, shear waves, and a longitudinal wave [33,42]. These generated waves were dependent on the density, modulus of elasticity, and Poisson's ratio of the material, as indicated

by Equation (1). Velocities from the three methods were then mathematically determined with Equations (2)–(4) [40,42,43]:

$$V_d \left( \frac{km}{s} \right) = \sqrt{\frac{1 - v}{(1 + v)(1 - 2v)} \frac{E}{\rho}}$$

(1)

$$V_{TSWV} \left( \frac{km}{s} \right) = \left( \frac{\pi D}{2} \right) \Big/ TOF$$

(2)

$$V_{LSWV} \left( \frac{km}{s} \right) = ab \Big/ TOF$$

(3)

$$V_{OSWV} \left( \frac{km}{s} \right) = \frac{\left( \frac{\pi D}{2} \right) + ab}{TOF}$$

(4)

where:

$V_d$ is the dilatational wave velocity,

$E$ is the dynamic modulus of elasticity and $\rho$ is the density of the wood,

$v$ is the Poisson's ratio of wood,

$D$ is the distance between the two probes—sending and receiving probes,

$TOF$ is the time taken for the stress wave to travel between probes.

$V_{TSWV}$, $V_{LSWV}$, and $V_{OSWV}$ are the sound wave velocities for transverse, longitudinal, and off-set methods, respectively. The $TOF$ is the time of flight recorded after tapping the sending probe five times and noticing readings that were within 0.02 microseconds of each other.

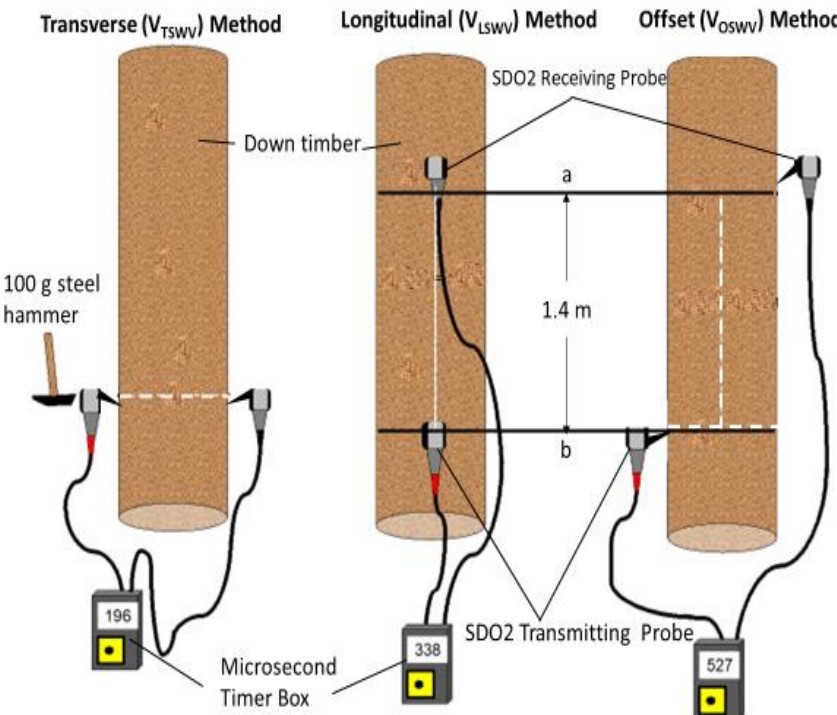

**Figure 1.** Measuring pattern using the FAKOPP acoustic tool; white dash lines indicate measured probe distance.

### 2.3. Assessing Fracture/Voids in Downed Timber

Acoustic tools were used to assess the downed timber internal fracture ends for bucking placement. This measurement was conducted at a 4 feet interval from the butt log and continued up to one-third the height of the tree. At every 4 feet, the star-pattern-shaped

measurement described in [44] was used. Any sharp changes to the tree's normal TOF could indicate fracture or decay. An internal fracture could disrupt the stress wave velocity (either abnormally high readings or low readings). Hence, the velocity from the butt or point of the snap was incrementally measured to the point of reasonable velocity (close to the velocity recorded at a point where there was no fracture or split on the same tree). The initial tree velocity of a standing tree was measured to determine the range, and a similar measure was conducted for a known fractured wood. The numbers obtained were used to predict the trees' hidden fractures. Downed trees picked by the acoustics to contain splits and voids were cut at the end of the project assessment to justify the acoustic readings.

*2.4. Determination of MC and Fiber Saturation Point on Downed Timber*

The increment borer was used to take five incremental samples at 10 ft intervals from the bark to the pith for each tree, starting at 4 ft. These samples were stored in plastic bags and placed in an ice cooler to maintain the moisture. The samples were taken to the lab, where the moisture content percentage (MC%) was estimated using the oven-dry method. The MC% was taken every two weeks, along with an acoustic measurement to determine when the log reached FSP. The core samples were oven-dried in batches at a temperature of 105 °C. The oven-dry weight of each sample was then measured and used in MC% estimation on dry basis as indicated by Equation (5):

$$MC\ (\%) = \frac{Green\ weight - Oven\ dry\ weight}{Oven\ dry\ weight} \times 100 \tag{5}$$

*2.5. Determination of Density*

Basic density is defined as oven-dry mass divided by fresh volume expressed in tons per cubic meter, kilograms per cubic meter, or grams per cubic centimeter [45,46]. The green density ($\rho_{green}$) and dry density ($\rho_{dry}$) of each tree was estimated from the freeze-dried cores, where the mass was taken at green weight ($W_d$) and dry weight ($W_{od}$), as shown in Equations (6) and (7):

$$\rho_{green}\left(\frac{g}{cm^3}\right) = \frac{w_g}{\pi\left(\frac{d}{2}\right)^2 h} \tag{6}$$

$$\rho_{dry}\left(\frac{g}{cm^3}\right) = \frac{w_{od}}{\pi\left(\frac{d}{2}\right)^2 h} \tag{7}$$

Treating the core as a cylinder, the volume of core ($V_c$) was estimated in Equation (8):

$$V_c\left(cm^3\right) = \pi\left(\frac{d}{2}\right)^2 h \tag{8}$$

where *d* represents the diameter, and *h* is the height of each core.

*2.6. Determination of Wood Stiffness*

The key parameter used to determine the structural grade of wood was stiffness, which is related to the dynamic modulus of elasticity $E_L$. The longitudinal stiffness and acoustic velocity in the longitudinal direction were related to a range of wood properties, such as the microfibril angle (MFA), [47,48] tracheid dimensions, and density [49]. The way a sound wave (stress wave) propagates through a body is correlated with the body's intrinsic properties. By measuring the velocity of the traveling wave and using the one-dimensional wave equation (Young's modulus equation), the dynamic modulus of elasticity was estimated as described by the fundamental one-dimensional wave equation, as shown in Equations (9) and (10):

$$V\left(\frac{km}{s}\right) = \sqrt{\frac{MOE}{\rho}} \tag{9}$$

$$MOE \text{ (MPa)} = E_L = V^2 \times \rho \tag{10}$$

where:

   $MOE$ or $E_L$ (MPa) stands for the modulus of elasticity,
   $V$ (Km/s) is the one-dimensional wave velocity,
   $\rho$ (kg/m$^3$) is the bulk density of the wood.

### *2.7. Data Analysis*

Several different statistical tests were employed to study the relationship between velocity and mechanical properties, as well as the deterioration rate, including simple linear regression and non-linear modeling. The data analysis was performed using SAS program (version 9.4, Cary, NC, USA) [50] and Minitab (version 19.0, State College, PA, USA) Statistical Software. The linear regression method was used to determine the level of the relationship between predictors and response variables. The response variables included the downed tree velocity and MOE. Meanwhile, age, dbh and dbh class, tree height, and density were considered the predictor variables. All the results from the acoustic velocity were normally distributed at ($p < 0.010$). An ANOVA (F-test for equality of means) and descriptive statistics were performed using the Minitab 19.0 software at an alpha level associated with a 95% confidence of 0.05 to test for the significant differences in stress wave velocities and MOEs. Factorial ANOVA and descriptive statistics were employed to assess the differences in the stress wave velocity and dynamic MOEs of the sample based on the predictor variables. The diameter classes were considered fixed variables ($Y_{ij}$) and tested for the relationship between the velocities and dynamic MOEs. The mean values and standard errors for the velocities were considered independent values ($B_j$), which were calculated using pivot tables with a designed model ($Y_{ijk} = \mu + \alpha_i + B_j + B_k + \varepsilon_{ij}$), where $\mu$ is the intercept and $\alpha_i$ is the assigned treatment such as downed timber characteristics (dbh, height, and density) and $\varepsilon_{ij}$ is the error assumed independent and normally distributed with mean 0 and variance $\sigma^2$. The post hoc analysis of Tukey's HSD test at a 95% confidence level was used to identify sample means that were revealed by ANOVA to be significantly different from each other. The generalized linear model was used to determine if the stress wave velocity could explain differences in downed timber stiffness and changes in moisture content. The models were evaluated at an alpha value of 0.05 ($\alpha = 0.05$).

### 3. Results

### *3.1. Downed Timber Physical Traits and Acoustic Velocity*

A summary of the downed timber characteristic average means, standard error of the means (SE mean), standard deviations (StDev), and coefficients of variations (CoV) analyzed in this study are presented in Table 1. The mean height and dbh of the chip-n-saw timber were 19.79 m and 0.24 m, while the sawtimber means were 23.15 m and 0.39 m, respectively. The green density among all trees did not differ much; however, the mean record for dry density was 0.47 g/cm$^3$ for the chip-n-saw and 0.511 g/cm$^3$ for the sawtimber, with coefficient variations of 11.31% and 12.98%, respectively. The mean acoustic velocity for the offset method VOSWV (5.09–5.56 km/s) was the highest for the three different types of measurement compared to the longitudinal VLSWV (3.33–3.63 km/s) method and the transverse method VTSWV (2.63–2.85 km/s). Similarly, the highest estimated dynamic MOE (EL) recorded was from the offset (ELVOSWV) method (12.46–16.079 MPa), with the lowest recorded by the transverse acoustic (ELVTSWV) methods (3.38–4.2298 MPa). The longitudinal (ELVLSWV) method recorded an MOE of 5.32 MPa with a standard error of the mean (SE mean) of 0.0247 and a standard deviation of 1.649 for the chip-n-saw, and an MOE of 6.89 MPa with a standard error of the mean (SE mean) of 0.0247 and standard deviation of 2.230 for the sawtimber. Generally, the sawtimber recorded the highest physical traits (height, dbh) and mechanical traits (density, velocity, and stiffness) compared to the chip-n-saw. The Person correlation between height and age was ~0.70 with a confident interval of 0.69, 0.71.

**Table 1.** Means, standard deviations and coefficient of variation (CoV) of downed timber traits by dbh specification class.

| ID | Chip-n-Saw (8″–11.9″) | | | | Sawtimber (12″ and Up) | | | |
|---|---|---|---|---|---|---|---|---|
| | Mean | SE Mean | StDev | CoV (%) | Mean | SE Mean | StDev | CoV (%) |
| h (m) | 19.786 | 0.0242 | 1.615 | 8.16 | 23.146 | 0.0316 | 2.857 | 12.34 |
| dbh (m) | 0.23787 | 0.000328 | 0.02183 | 9.18 | 0.39225 | 0.000807 | 0.07287 | 18.58 |
| P green (g/cm$^3$) | 0.86372 | 0.00121 | 0.08049 | 9.32 | 0.83383 | 0.000898 | 0.0811 | 9.73 |
| P dry (g/cm$^3$) | 0.46831 | 0.000795 | 0.05296 | 11.31 | 0.51153 | 0.00073 | 0.06595 | 12.89 |
| V$_{TSWV}$ (km/s) | 2.633 | 0.00744 | 0.4957 | 18.83 | 2.8475 | 0.00404 | 0.3647 | 12.81 |
| V$_{LSWV}$ (km/s) | 3.3327 | 0.00658 | 0.4382 | 13.15 | 3.6283 | 0.00555 | 0.5015 | 13.82 |
| V$_{OSWV}$ (km/s) | 5.0896 | 0.011 | 0.7348 | 14.44 | 5.5633 | 0.00661 | 0.5974 | 10.74 |
| EL$_{VTSWV}$ (MPa) | 3.3794 | 0.0187 | 1.2469 | 36.9 | 4.2298 | 0.014 | 1.2652 | 29.91 |
| EL$_{VLSWV}$ (MPa) | 5.3204 | 0.0247 | 1.6485 | 30.98 | 6.8888 | 0.0247 | 2.2304 | 32.38 |
| EL$_{VOSWV}$ (MPa) | 12.46 | 0.0602 | 4.011 | 32.19 | 16.079 | 0.0463 | 4.179 | 25.99 |

### 3.2. Acoustic Velocity, Age, and Diameter Class

The effects of acoustic velocity on the inter-relationships between the measures of wood quality were examined. The samples were first grouped by age and by dbh classes. Because the dbh class overlapped across the age categories, it was used to determine whether there was a significant effect of the dbh class on the different acoustic measures (Figure 2). For the transverse, longitudinal, and offset methods, the sawtimber with a dbh class of 12″ and up recorded a higher acoustic velocity than the chip-n-saw. The chip-n-saw diameter class was between 8″ and 11.9″, as described by Timber Mart South. The sawtimber for the transverse acoustic velocity (VTSWV) was 3.91% higher than chip-n-saw. In comparison, the longitudinal method (VLSWV) was 4.25% higher. The offset method (VOSWV) recorded a 4.45% higher acoustic velocity for the sawtimber than the chip-n-saw. Using the Tukey pairwise method and a 95% confidence grouping, all the differences between the acoustic velocity measurement for the sawtimber and chip-n-saw were significant ($p \leq 0.001$).

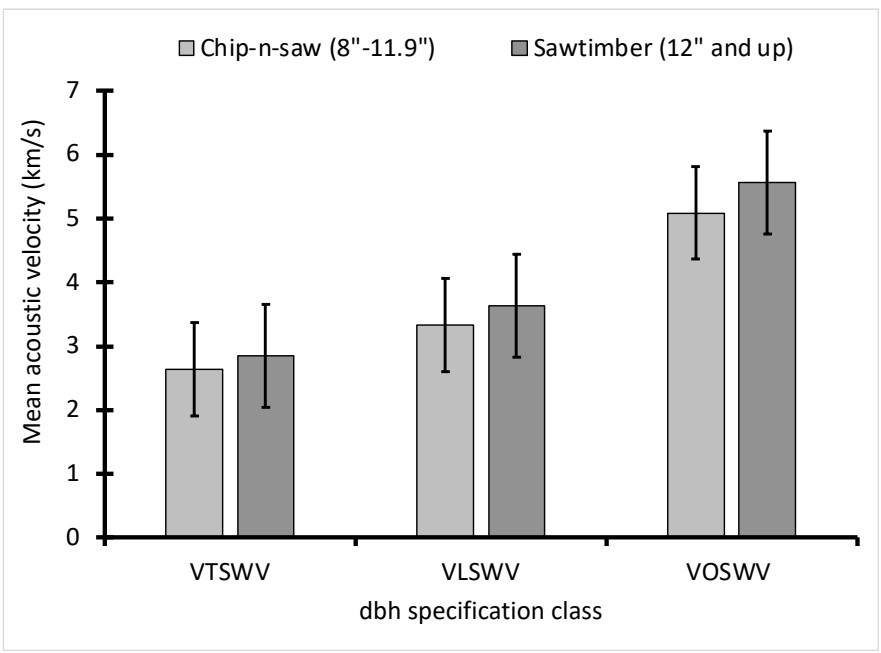

**Figure 2.** Different acoustic velocity measuring types and dbh specification class.

The performance of the different acoustic measurements with the age of the downed timber is shown in Figure 3. The older trees recorded a higher acoustic velocity than the younger trees, and the offset method also recorded higher acoustic velocity than the

longitudinal and transverse methods. The lowest record was attained by the transverse method across the age categories. The downed tree age and dbh classes were significant ($p \leq 0.001$). The nested analysis of variance indicated that the coefficient of the dbh class was lower than the ages of the tree. However, the standard error of the coefficient of the chip-n-saw and the trees aged 30 years were almost the same (0.016) (Table A1).

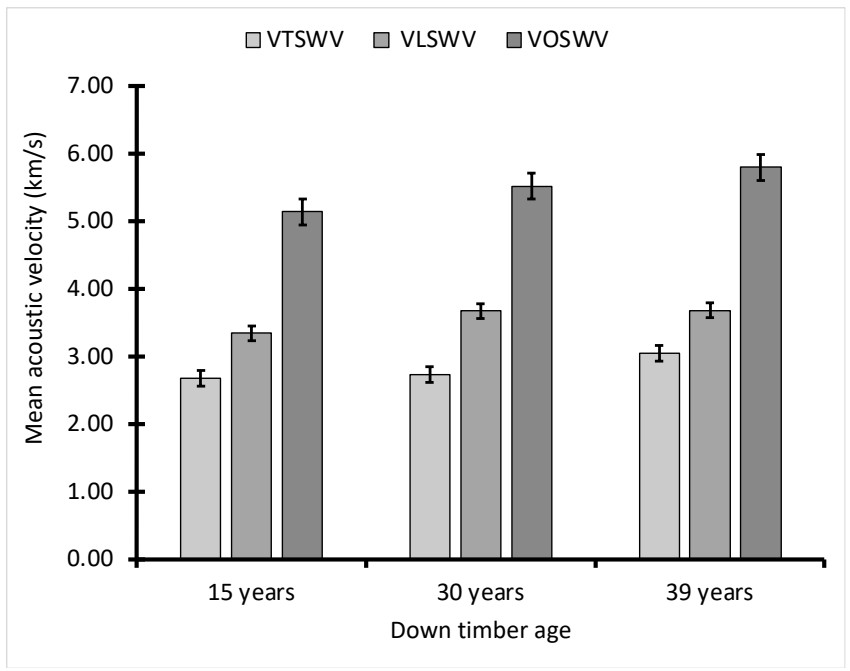

**Figure 3.** Different acoustic velocity measuring types and ages of downed timber.

### 3.3. Mechanical Properties and Acoustic Velocity

This study showed a strong association of the downed trees' stiffness (dynamic MOE) with acoustic velocity for all the three different methods, as shown above (85%; Figure 4). Furthermore, the regression analysis indicated a strong positive correlation for transverse acoustic ($R^2 = 0.86$), longitudinal acoustic ($R^2 = 0.87$), and offset acoustic methods ($R^2 = 0.86$) between the acoustic velocity and their respective dynamic MOEs.

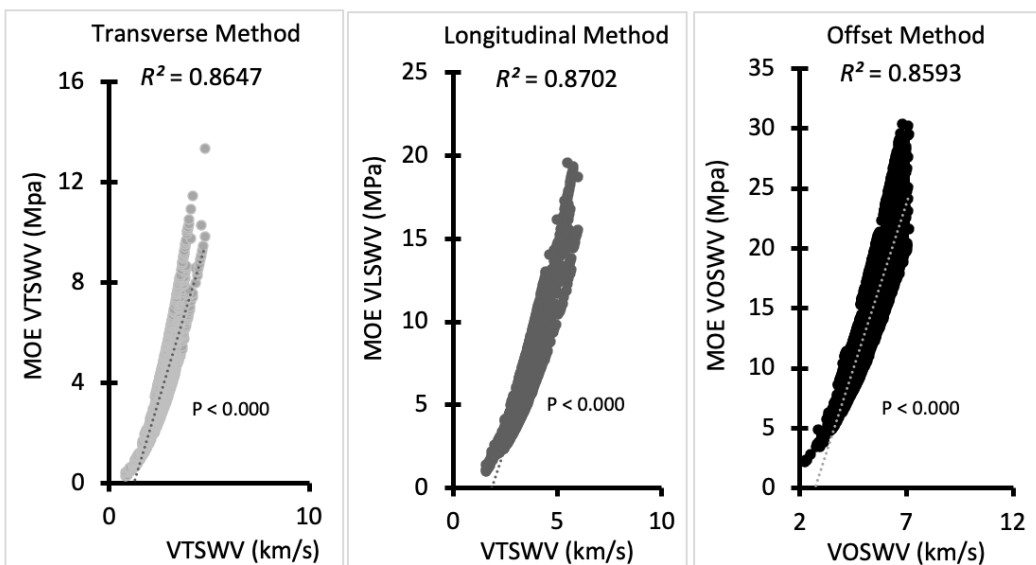

**Figure 4.** Different acoustic velocity measuring types and dynamic modulus of elasticity.

### 3.4. Precipitation, TOF, and Acoustics

Figure 5a and b show the monthly precipitation at the site and the mean moisture content (MC) in percentages of the downed trees taken from the core samples, respectively. The moisture content at one year did not reach the fiber saturation point (~30% MC), as shown in Figure 5b, by the project completion time. The monthly precipitation pattern followed a similar trend to that shown in the MC readings (Figure 5a). The moisture readings for the downed trees were concurrently taken with the time of flight (TOF) acoustics on a biweekly and, subsequently, a monthly basis. The acoustics results showed a gradual increase in the TOF results for all tree measurements, with the offset method excluded in this study (Table 2). The gradual rise in TOF caused a decrease in stiffness which could be attributed to a change in the chemical composition of the wood or wood deterioration. However, because the moisture content was high, the difference in the acoustics was not rapid, implying that the decay/decomposition was slow. The acoustics subtly identified this decomposition as a gradual increase in TOF for both the longitudinal and transverse acoustic measurements. The increase in TOF caused a decrease in acoustic velocity and stiffness. Decay or deterioration was not noticeable since the acoustics did not notably change above the fiber saturation point. The acoustic velocity from week 1 to week 49 showed no significant changes in the downed timber acoustic velocity (171 km/s–187 km/s) except in week 11, which increased to 200 km/s for the transverse method. Comparing the data in week 11 to the MC% and NOAA precipitation, reports showed a relative decrease in moisture percentage and precipitation. The value of the precipitation, MC%, followed a similar hailstone sequence shown in the acoustic TOF for certain months, which was dictated by changes in the weather, particularly precipitation. The study found that some of the water wicks into the wood when it rains, especially in the butt region, which is exposed without bark covering, and to some extent into the outer wood, which affects the acoustic reading.

**Table 2.** Time of flight (TOF) acoustic measurements for transverse and longitudinal methods.

| Weeks | Transverse Method | | | | | Longitudinal Method | | | | |
|---|---|---|---|---|---|---|---|---|---|---|
| | Mean (µs) | Min (µs) | Max (µs) | StdDev | CoV (%) | Mean (µs) | Min (µs) | Max (µs) | StdDev | CoV (%) |
| 1 | 171 | 86 | 265 | 40 | 23.10 | 422 | 255 | 618 | 50 | 11.80 |
| 7 | 194 | 110 | 298 | 45 | 23.21 | 396 | 255 | 504 | 41 | 10.47 |
| 9 | 185 | 100 | 292 | 50 | 27.15 | 412 | 244 | 769 | 87 | 21.26 |
| 11 | 200 | 114 | 316 | 49 | 24.36 | 398 | 264 | 528 | 40 | 10.03 |
| 13 | 190 | 105 | 297 | 50 | 26.60 | 416 | 249 | 774 | 87 | 20.82 |
| 15 | 178 | 102 | 366 | 49 | 27.60 | 368 | 234 | 825 | 55 | 15.05 |
| 17 | 175 | 93 | 310 | 48 | 27.23 | 381 | 247 | 578 | 44 | 11.64 |
| 19 | 179 | 92 | 326 | 50 | 27.86 | 391 | 243 | 853 | 53 | 13.52 |
| 21 | 183 | 100 | 397 | 51 | 27.90 | 397 | 235 | 884 | 55 | 13.92 |
| 23 | 188 | 102 | 368 | 52 | 27.58 | 402 | 247 | 554 | 47 | 11.75 |
| 25 | 189 | 102 | 343 | 52 | 27.60 | 401 | 259 | 542 | 48 | 12.06 |
| 27 | 187 | 108 | 384 | 45 | 24.21 | 415 | 253 | 805 | 53 | 12.81 |
| 29 | 218 | 98 | 552 | 52 | 23.85 | 411 | 252 | 876 | 72 | 17.63 |
| 34 | 216 | 109 | 368 | 44 | 20.13 | 410 | 259 | 855 | 61 | 14.91 |
| 43 | 216 | 128 | 374 | 45 | 20.84 | 443 | 288 | 879 | 63 | 14.25 |
| 49 | 226 | 135 | 505 | 65 | 28.76 | 427 | 237 | 708 | 67 | 15.59 |
| Grand Total | 194 | 86 | 552 | 52 | 27.09 | 406 | 234 | 884 | 61 | 15.03 |

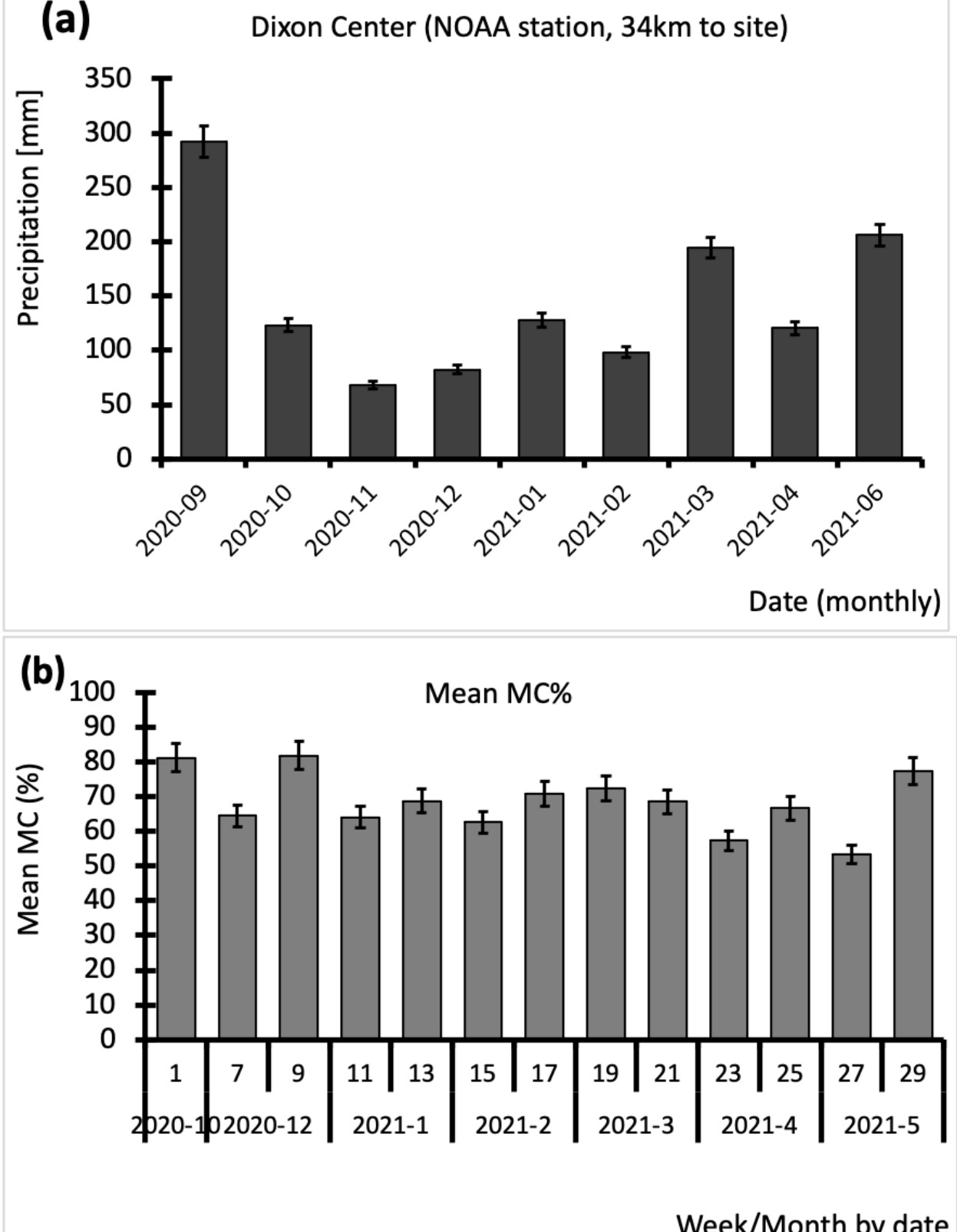

**Figure 5.** NOAA monthly precipitation at the site, (**a**) and the mean moisture content (MC) in percentages of the downed trees (**b**).

### 3.5. Deterioration and Void Detection

Visual inspection was conducted around the butts of the trees for damage assessment, including snaps, mechanical damage, deep checks or splits, decay, and signs of actual or potential deterioration. In addition, key indicators such as fruiting bodies, staining, water damage, evidence of insect activity (holes, frass, and powder) were assessed as indicators of deterioration or damage (Figure 6). Some blue stains and insect damage were seen around the first three months, and bark peeling was observed in some downed trees within the first month. Woodpeckers were seen on sight during the second week of the site visit. However, woodpecker damage to the downed trees is not very harmful; it only creates wounds where diseases and insects can enter the tree.

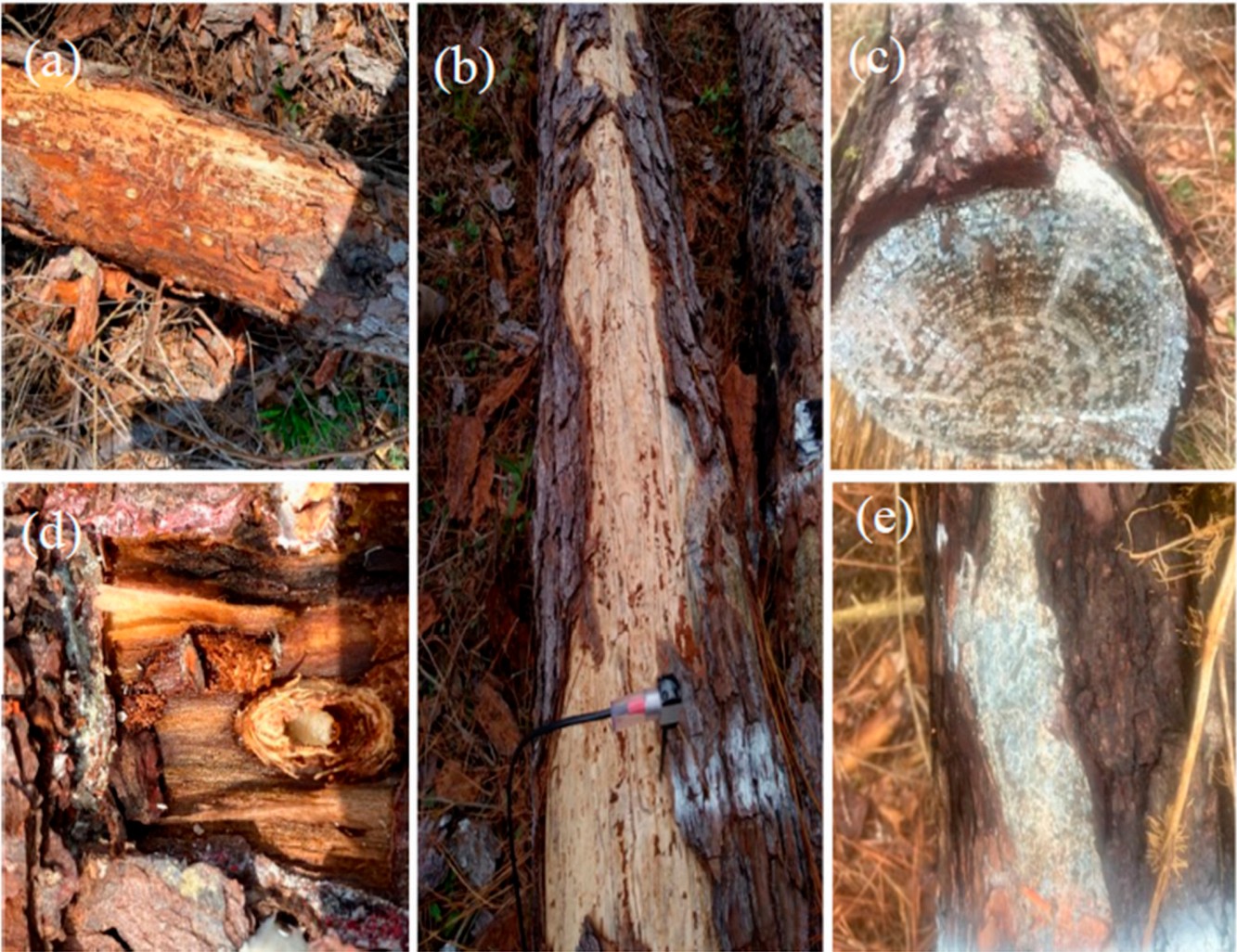

**Figure 6.** Visual assessment of downed trees for signs of decay and damage (**a**) woodpecker holes and insect damage, (**b**) peeling of the bark, (**c**) blue stain, (**d**) insect eggs and pupa and (**e**) stains and frass.

The TOF acoustic measures for splits and voids for the 30-year-old tree (1990) are shown in Figure 7. The trees with extremely high acoustic regions from the normal acoustic response of the trees above dbh for the transverse star pattern (>250 µs) were considered to have split damage. The analysis of the results indicated that TOF increases with voids, checks, and splits. TOF is inversely proportional to acoustic velocity—implying that if TOF increases, the acoustic velocity decreases. Most voids from downed timber seemed to be limited to the first 150 cm of the point of damage (this value was located higher in storm-damaged trees). The undamaged trees were seen to have consistent TOF (not

significant, *p* < 0.54) along the tree length, although this slightly decreased along the slender length of the downed tree (1990, Tree 1).

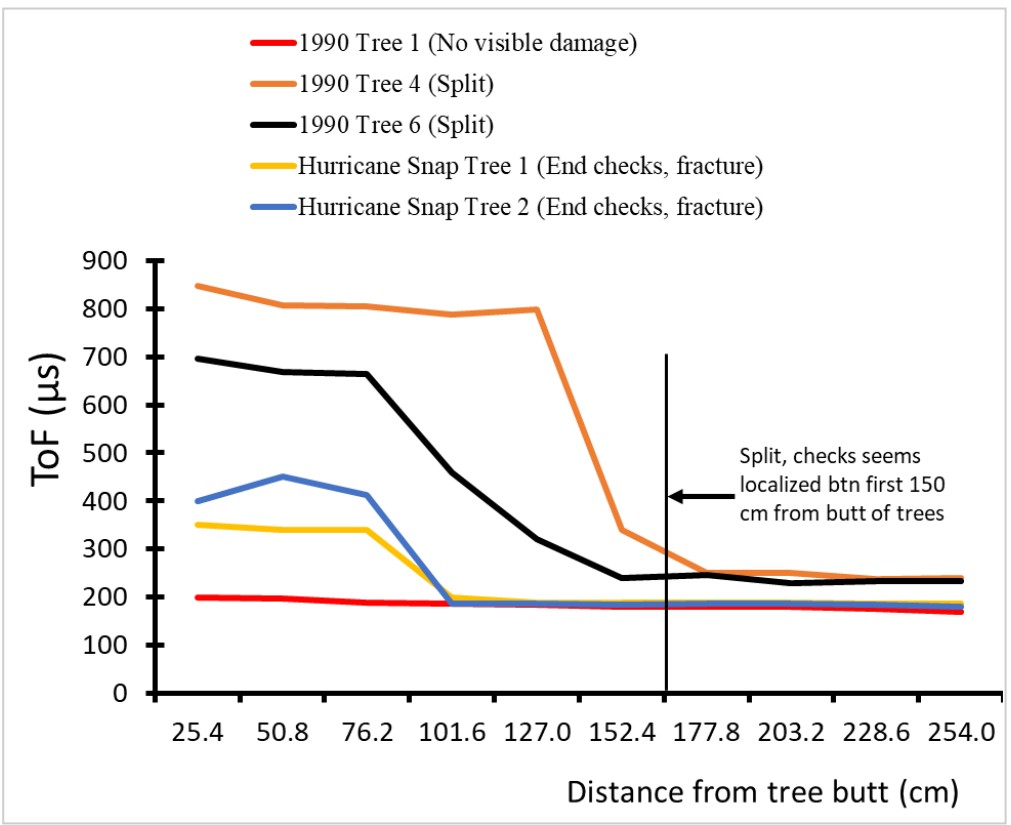

**Figure 7.** Internal damage of downed trees picked by acoustic readings from the butt of the tree.

## 4. Discussion

The acoustic velocity of a stress wave generated by a mechanical impact offers reliable opportunities to evaluate standing trees and logs for their general wood quality and intrinsic wood properties [33,51]. This technology was extended to downed trees in this study with a particular interest in storm-damage applications. Field data from downed loblolly pine trees from three different sites with different dbh and stand ages were used to examine the acoustic velocities of trees, as measured by TOF tools. The analysis was based on the timber specification dbh classes for the southeastern region of the United States [37]. Even though the allometric relationship between stem diameter at breast height and tree height does not always represent stem-mass growth patterns [52], the study shows that the more mature trees with higher dbh (sawlogs) have relatively higher tree diameters and heights (Table 1). Tree height is an essential feature in scaling forest quantities [53] With the appropriate application of laws, the size of the tree is a good indicator of some of its dynamic structure and physiological traits [54]. The Pearson correlation between height and age was ~0.70 with a confidence interval of 0.69, 0.71. Other studies reported a significant correlation of 0.42 (*p* = 0.02) for small clear wood samples collected from 183 Douglas fir trees (velocities versus air-dried density) [55]. Comparing the green density to the dry density in the study indicated that density was highly influenced by the weight of the moisture [44]. Consequently, a reduction in moisture by the oven-dry method led to a significant decrease in the dry density compared to the green density for both the chip-n-saw and the sawlogs. Density is one of the consistently important drivers of stiffness and velocity [56], and it is especially important to consider when studying downed trees. The importance of height, age, and dbh observations is key for determining the use of downed timber, particularly small diameter sawtimber. Its application based on density and stiffness in sawmills provides several diverse functions, the major factor driving forest

investment. The offset method recorded a relatively higher acoustic velocity and dynamic MOE than the longitudinal method, and the transverse method recorded the lowest velocity and stiffness performance. These results align with the study by Mahon et al. [41], who compared different sampling methods for tree acoustic measurement. The study found that transmitting and receiving probes placed on the same and opposite faces showed significant velocity differences. The velocity determined using the circumferential opposite-face method was higher and depended on stem size or the amount of wood through which the stress wave must pass.

The acoustic velocity showed a positive linear relationship with density as indicated by these results and conformed to several previously reported studies [28,40,57,58]. Other studies showed a trend of decreasing acoustic velocity versus dbh at the individual tree level [51,59]. Age, dbh, and silvicultural operations were often indicated to influence the relationship of velocity [40,60,61], where the nested analysis of variance indicated that both age and dbh were significant contributors to the velocity and stiffness performance. The chip-n-saw timber generally had lower acoustics and stiffness, with the possibility of a high longitudinal shrinkage making them questionable for use as lumber for production as structural materials [62]. However, due to the low prices of chip-n-saw as a small diameter timber [63], particularly in the case of low-grade downed timber, value-added products such as CLT need to be explored [64], so that harvesting can be economically feasible [65]. The chip-n-saw may be repurposed for non-structural application materials, such as OSB, particleboard, or uses in plastic composites [66,67]. The strong association of a downed tree's stiffness with acoustic velocity for all methods is indicated by the linear regression to be above 85%. This outcome aligns with several studies that showed a strong relationship between acoustic speeds and mechanical properties [27,61,68,69]. The attained difference between the different measurements for the three different methods could be attributed to several factors, including the travel path [41], density, age, and microfibril angle [57,68]. Understanding the stiffness of downed timber based on the modulus of elasticity [62] is especially important as a component for determining the mechanical properties of wood and engineered wood products [70].

The results from the 12-month acoustic response were not significantly different for the transverse method ($p \leq 0.045$) and for the longitudinal method ($p \leq 0.116$) above the fiber saturation point (Figure 5). This is in line with the studies by [71–73], which indicated that, above FPS, changes in moisture has an insignificant effect on velocity. Comparing the moisture content reading to the NOAA precipitation [74] reading for the study site with the acoustic performance in Table 2, it can be said that the gradual variation changes across the weeks and months were mainly due to the weather and changes in chemical compositions. The lower variability of the acoustic performance in the first 12 months for individual trees was attributed to the moisture content, which was low. Log air drying was very low and could take up to 3 years for some species to reduce below FPS [75]. The prolonged drying period of logs is dictated by several factors, including their density, bulkiness, bark thickness, environmental temperature, and humidity, among many other factors. Previous studies indicated that acoustics could predict whole-log velocity and dynamic MOE [3]; however, the outer wood zone affected TOF readings, [76] as shown in this study. This implies that landowners and scientists, when assessing the quality of the wood in the field using acoustic tools, should be careful when interpreting data obtained from various parts along and across the tree over different edaphic and environmental conditions. Studies have shown that, above the fiber saturation point, velocity decreases by less than 2% per unit increase in moisture content, while the density exponentially increases per unit increase in moisture content [73,77].

A visual assessment aimed to estimate the overall health of the downed trees [78–80]. However, it is very difficult to accurately predict the extent of decay and hollow in trees with a visual assessment, and the use of intrusive tools, such as incremental core borers, shigometer, or resistograph [81–83], comes with several drawbacks. The slow decay process of the downed tree from the acoustic measurements, aligns with the visual inspections

conducted at each site visit. The decrease in TOF along its length indicates a decrease in stiffness, which is confirmed by previous studies where a significant positive relationship existed between standing tree stiffness and stem slenderness [69,84]. The downed trees identified by the acoustic tools to have voids and splits were cut with a chainsaw following the length measurement at the end of the project, and the resulting damage revealed by the cuts was consistent with the prediction of the acoustics, as shown in Figure 7. Internal voids and fracture are hard to see and are difficult to predict, and this process can result in hurricane snap regions used for bucking decisions; using this pilot study gives an idea of bucking decisions as dictated by acoustics. Foresters and landowners are often quick to sort pieces close to breaking into lower-quality product categories due to the uncertainty of how far the damage extends into wood that appears to be solid; the use of the TOF by acoustics reduces this challenge to some extent. Similar approaches can be employed in any storm-damage conditions for bucking decisions. The use of the nondestructive evaluation method, which aims to detect splits, voids, and decay in the downed trees to help landowners and forest managers identify hazardous downed trees, prevents the spread of decay and improves salvaging decisions. The TOF acoustic method is nonintrusive for detecting the presence of voids and decay with minimal damage to the tree [78].

## 5. Conclusions

The results of this study indicate that acoustic velocity measurements using the time-of-flight method give a good indication of downed tree velocity and stiffness, which can be employed in hurricane events as NDE techniques to confirm the relationship between acoustic speeds and the mechanical properties of timber species. Additionally, based on the linear regression results of the dynamic MOE, the acoustic velocity can be used to separate higher-stiffness timber from lower-stiffness timber in a hurricane event for structural or non-structural applications. Chip-n-saw timber generally has lower acoustics and stiffness with the possibility of high longitudinal shrinkage, making it questionable for use as lumber in the production of specific structural materials. However, they may be repurposed for materials used for non-structural applications, such as OSB and particleboard, or used for plastic composites.

The results of this research, based on the visual assessment and gradual change in TOF acoustic measures tailored to decay indicators, show that a careful observation of symptoms by landowners and foresters paired with the use of acoustics in a storm-damaged tree could lead to fairly accurate predictions of internal tree decay. Based on the acoustic results, we could eliminate current impediments that discouraged the use of downed timber following catastrophic weather events, such as unseen voids and particularly those close to breaking, which are often classified in lower quality product categories. This study suggests that landowners and log graders should use acoustics tools (Fakkop Microsecond timer) to obtain a comprehensive idea of the condition of storm-damaged trees to verify and recommend their possible applications. However, various factors, particularly edaphic and environmental conditions, could significantly affect the acoustic velocity and dynamic MOEs obtained, indicating that caution should be an exercise in interpreting results in such situations. Further research is suggested for applying this tool to different storm-damaged trees, particularly those damaged by hurricanes, to improve the accuracy and power of this method for the possible of prescreening hurricane- and tornado-damaged timber.

**Author Contributions:** Conceptualization, B.V.; methodology, M.M., B.V., T.G., M.S.P., D.M. and M.S.; validation, M.M., B.V., D.M., M.S. and M.S.P.; formal analysis, B.V. and M.M.; investigation, M.M., A.O.A. and J.H.D.; resources, T.G., D.M., M.S. and B.V.; data collection, M.M., J.H.D. and A.O.A.; data curation, M.M. and B.V.; writing—original draft preparation, M.M. and B.V.; writing—review and editing, B.V., A.O.A., M.S., J.H.D. and M.S.P.; visualization, M.M. and J.H.D.; supervision, B.V.; project administration, B.V., D.M., M.S., T.G. and M.S.P.; funding acquisition, B.V., D.M., M.S., T.G. and M.S.P. All authors have read and agreed to the published version of the manuscript.

**Funding:** This research was funded by the USDA Forest Service Down Timber Research Fs-20-Ca-11330170-066-Bv, the McIntire-Stennis (MS) Cooperative Forestry Research Program and the Hatch Project.

**Data Availability Statement:** The NOAA precipitation and weather report for study area can be found at https://www.ncdc.noaa.gov/cdo-web/ (accessed on 6 April 2022).

**Conflicts of Interest:** The authors declare no conflict of interest. The funders had no role in the design of the study; in the collection, analyses, or interpretation of data; in the writing of the manuscript, or in the decision to published results.

## Appendix A

**Table A1.** The coefficients of age and dbh class.

| Term | Coef | SE Coef | T-Value | *p*-Value | VIF |
|---|---|---|---|---|---|
| Constant | 4.1449 | 0.0133 | 310.50 | 0.000 | |
| dbh class | | | | | |
| Chip-n-saw | −0.1269 | 0.0162 | −7.82 | 0.000 | 2.50 |
| Age (yrs) | | | | | |
| 15 | −0.6386 | 0.0208 | −30.63 | 0.000 | 2.64 |
| 30 | −0.4897 | 0.0165 | −29.74 | 0.000 | 1.42 |

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
