# Peer review of "Field Assessment of Downed Timber Strength Deterioration Rate and Wood Quality Using Acoustic Technologies"

_forests, doi:10.3390/f13050752_

Round 1

Reviewer 1 Report

In this paper, the authors used acoustic wave techniques as a non-destructive testing method to assess the degree of wood degradation in felled trees and determine the extent of cracks and voids in damaged areas of trees. The topic is interesting and important because hurricanes and tornadoes destroy significant areas of forest every year, leaving a damaged stand of trees that is unlikely to have structural uses. Perhaps not so much because it will undoubtedly show inferior properties, but such use of broken trees is avoided for safety reasons. The authors suggest using a non-invasive method to assess the degree of wood degradation, which will eliminate these trees from structural use at the harvest site. And leave those that have not been damaged. And can be used in structural elements despite being knocked down by winds.
Methodologically, the work was done correctly. A very detailed description of the research methods used has been made. The discussion of the results is also thorough and multi-threaded. 
The description of equations 1-9 (the units should be given) and the description of Figure 6 require minor corrections.

Author Response

Dear Dr. Mila Gao,

Thanks a lot indeed for the time in reading the manuscript and for the excellent comments on the paper. In the new version all changes are highlighted in red. As suggested by the reviewers. We made changes in some parts of methodology and in the results.

All changes are explained in detail in the response to the reviewer’s comments as indicated

below.

Thanks a lot again for your very helpful and constructive comments on the paper.

Best regards,

Munkaila Musah, PhD.

Reviewer 2: The description of equations 1-9 (the units should be given) and the description of Figure 6 require minor corrections.

Response:

The units have been added to equation 1-10. Each equation has its unit in a bracket and the alphabets for figure 6 that represent each figure has been updated accordingly (b. was repeated twice instead of b. and c.)

Reviewer 2: In general, the manuscript “Field assessment of downed timber strength deterioration rate and wood quality using acoustic technologies” presents an understandable text and current references in the discussion, with no need for further corrections.

Response:

Well noted and thank you

Reviewer 2 Report

In general, the manuscript “Field assessment of downed timber strength deterioration rate and wood quality using acoustic technologies” presents an understandable text and current references in the discussion, with no need for further corrections.

Author Response

Dear Dr. Mila Gao,

Thanks a lot indeed for the time in reading the manuscript and excellent comments on the paper. In the new version all changes are highlighted in red. As suggested by the reviewers we made changes in some parts of methodology and in the results.

All changes are explained in detail in the response to the reviewer’s comments as indicated

below.

Thanks a lot again for your very helpful and constructive comments on the paper.

Best regards,

Munkaila Musah, PhD.
